# Effect of Prior ChAdOx1 COVID-19 Immunisation on T-Cell Responses to ChAdOx1-HBV [note 1]

**DOI:** 10.3390/vaccines12060644

**Published:** 2024-06-09

**Authors:** Charlotte Davis, Dave Singh, Katie Anderson, Antonella Vardeu, Jakub Kopycinski, Alice Bridges-Webb, Alice Trickett, Susanne O’Brien, Matthew Downs, Randip Kaur, Radka Kolenovska, Louise Bussey, Kathryn Rutkowski, Sarah Sebastian, Tamsin Cargill, Eleanor Barnes, Thomas G. Evans, Paola Cicconi

**Affiliations:** 1Barinthus Biotherapeutics, Harwell, Didcot OX11 0DF, UKantonella.vardeu@barinthusbio.com (A.V.); matthew.downs@barinthusbio.com (M.D.);; 2Medicines Evaluation Unit Ltd., Manchester M23 9QZ, UK; dsingh@meu.org.uk; 3Nuffield Department of Medicine, University of Oxford, Oxford OX1 2JD, UKellie.barnes@ndm.ox.ac.uk (E.B.); 4Centre for Clinical Vaccinology and Tropical Medicine (CCVTM), University of Oxford, Oxford OX1 2JD, UK; paola.cicconi@ndm.ox.ac.uk

**Keywords:** hepatitis B, vaccine, T cell, Vaxzevria

## Abstract

There are varying data concerning the effect of prior anti-vector immunity on the T-cell response induced by immunisation with an identical vectored vaccine containing a heterologous antigen insert. To determine whether prior exposure to ChAdOx1-SARS-CoV2 immunisation (Vaxzevria^®^) impacts magnitudes of antigen-specific T-cell responses elicited by subsequent administration of the same viral vector (encoding HBV antigens, ChAdOx1-HBV), healthy volunteers that had received Vaxzevria^®^ (n = 15) or the Pfizer or Moderna mRNA COVID-19 vaccine (n = 11) between 10 and 18 weeks prior were recruited to receive a single intramuscular injection of ChAdOx1-HBV. Anti-ChAdOx1-neutralising antibody titers were determined, and vector or insert-specific T-cell responses were measured by a gamma-interferon ELISpot and intracellular cytokine staining (ICS) assay using multiparameter flow cytometry. Participants were followed for three months after the ChAdOx1-HBV injection, which was well-tolerated, and no dropouts occurred. The baseline ChAdOx1 neutralisation titers were higher in the Vaxzevria^®^ cohort (median of 848) than in the mRNA cohort (median of 25). T-cell responses to HBV antigens, measured by ELISpot, were higher on day 28 in the mRNA group (*p* = 0.013) but were similar between groups on day 84 (*p* = 0.441). By ICS, these differences persisted at the last time point. There was no clear correlation between the baseline responses to the adenoviral hexon and the subsequent ELISpot responses. As vaccination within 3 months using the same viral vector backbone affected the insert-specific T-cell responses, a greater interval after prior adenoviral immunisation using heterologous antigens may be warranted in settings in which these cells play critical roles.

## 1. Introduction

There are concerns that the prior use of low seroprevalence viral-vectored vaccines may result in the induction of anti-vector responses, which would decrease the response to a subsequent use of the same vector. Specifically, there have been concerns raised that the widespread use of the AstraZeneca ChAdOx1-SARS-CoV-2 vaccine (AZD1222 or Vaxzevria^®^) might reduce either the antibody or T-cell response to a subsequent vaccine using the same ChAdOx1 vector or other adenoviral vectors. This hypothesised decrease has been attributed to either neutralising antibodies clearing the virus prior to transfection or to the induction of a cross-reactive T-cell response to the viral vector that more rapidly eliminates the antigen-presenting cells.

For adenoviral vectors, prior infection with a natural replicating adenovirus of the same strain (e.g., human adenovirus 5) appears to influence the subsequent T-cell and antibody responses [1,2,3,4]. In such instances, neutralising responses are primarily directed against the fibre protein [5]. The seroprevalence resulting from prior infection driving this interference was a primary motivating factor in the development of replication-incompetent adenoviral vectors based on simian strains (chimpanzee, bonobo, and gorilla) [6]. In contrast, neutralisation, resulting from the use of replication-incompetent adenoviral vectors themselves, is directed primarily towards the abundant hexon capsid protein, and this response may only have a minor effect on subsequent cell entry and immunogenicity, mainly mediated by the fibre [5].

In studies of AZD1222, there was no statistical relation between the level of adenoviral neutralisation titers following the first immunisation and the resultant antibody or T-cell response (as measured by interferon-gamma (IFN-γ) ELISpot) following a second immunisation [7,8]. Furthermore, administration of a ChAdOx1 vaccine (either ChAdOx1-MERs or ChAdOx1-mening) over one year prior had no effect on the antibody response to AZD1222 [9].

However, an improvement in antigen-specific immunogenicity using a delayed interval is supported by reports in the literature of clinical trials assessing homologous or heterologous prime-boost regimes with adenoviral vectors [10,11,12,13,14]. In most cases, delaying the interval between a first and second dose of the adenoviral vector improves the antigen-specific boosting capacity of the second dose. This has been mostly studied with antibody responses to the same adenoviral vector encoding the same antigen and has been attributed in part to the declining titers of vector-neutralising and vector-binding antibodies with time. Specifically, antigen-specific antibodies were boosted 3-fold when the boosting interval was 4 weeks, but this increased to 10-fold when the interval between the two vaccines was increased to 24 weeks [11]. In another study, antigen-specific antibody titers were boosted 10-fold after a prime-boost interval of 12 weeks, and the T cells were also boosted [10]. Nonetheless, it is well known that simply increasing the interval may boost responses. However, the association of the increased antibody titers with delayed intervals has not been clearly demonstrated in any of the cited studies to be directly attributable to the level of anti-vector-neutralising antibodies [15].

It is important to note that most previous clinical experience with a second administration of the same adenoviral vector has been in the context of a single antigen, for example, Ad26 prime followed by Ad26- boost [10,14] or heterologous vector backbones [16,17]. This contrasts with the scenario discussed here, where a possible administration of the ChAdOx1-nCoV-19 vaccine is followed by the administration of a ChAdOx1 vector encoding HBV antigens. This is not a classic prime-boost scenario per se but rather the re-use of the same vector platform in a different indication.

Given this uncertainty and the improved responses seen three months after initial immunisation compared to one month in studies of the homologous prime-boost of the same antigen, and lacking clear data, Barinthus Biotherapeutics-sponsored trials employing the ChAdOx1 vector recommended at least a three-month interval after receiving the AZD1222 (Vaxzevria^®^) or J&J hAd26. However, this was not based on firm data, and either a shorter or longer interval may indeed be warranted.

To support this interval, we studied the responses to our ChAdOx1-HBV immunotherapeutic in healthy volunteers following the use of the AZD1222 vaccine. ChAdOx1-HBV is the initial dose of Barinthus Biotherapeutic’s antigen-specific immunotherapy (VTP-300), which encodes multiple hepatitis B antigens, including full-length surface, modified polymerase and core antigens. T-cell responses (the critical immunological factor in chronic HBV infection) were compared in participants who received either a prior two-dose series of AZD1222 with those who have received at least two prior doses of the Pfizer/Moderna mRNA COVID-19 vaccine (Comirnaty^®^/Spikevax), to determine whether prior exposure to AZD1222 impacts the magnitudes of antigen-specific T-cell responses elicited by subsequent administration of the same viral vector (ChAdOx1-HBV).

## 2. Materials and Methods

Participants: The study was conducted as an amendment to the published Phase 1 study of ChAdOx1-HBV [18]. The study followed all the ethical considerations listed in that manuscript. Cohort 5 (prior ChAdOx1, referred to as PC, n = 15) consisted of healthy adult males or females aged ≥40 to ≤60 years at screening who had completed a second dose of COVID-19 AZD1222 vaccine 10 to 18 weeks before enrolment. Cohort 6 used the same age group who had received the latest dose of either Pfizer (Comirnaty^®^) or Moderna (Spikevax) mRNA COVID-19 vaccine, 6 to 30 weeks before enrolment (PM, n = 11). T-cell assessments were carried out using cryopreserved PBMC samples collected on days 0, 14, 28, and 84. In addition, neutralising antibodies to ChAdOx1 were assessed using frozen serum samples collected on days 0 and 84. The patients were recruited and managed at a single site (Medicine Evaluation Unit Ltd., Manchester, UK), and PBMC isolation and cryopreservation were carried out at a nearby laboratory (GCP Laboratory Facility, University of Liverpool, Liverpool, UK).

Endpoints: The primary endpoint was to assess whether the receipt of prior ChAdOx1-SARS-CoV-2 vaccine results in decreased T-cell responses to ChAdOx1-HBV when administered 10–18 weeks prior to ChAdOx1-HBV, as measured by a peptide-stimulated IFNγ ELISpot assay. The secondary endpoints included safety and the CD4+ and CD8+ T-cell magnitudes and phenotypes, as measured by multiparameter flow cytometry.

IFNγ ELISpot: The IFNγ ELISpot was used to detect the frequency of HBV antigen or hexon-specific T cells in PBMC across the study’s time points. Ninety-six well PVDF plates (Millipore, Watford, UK) were coated overnight at 4 °C with 4 µg/mL of anti-human IFNγ capture antibody (BD Biosciences, Wokingham, UK). The plates were washed and blocked with Complete Media for 2 h. The plates were then loaded with 2 µg/peptide/mL of 15-mer peptides (overlapping by 11 amino acids) covering the HBV core, polymerase (Pol), and surface HBV antigen sequences (Mimotopes), arranged into four pools (Core, Polymerase 1+2, Polymerase 3+4, Pre-S1+S2 and Surface) or AdV5 Hexon (PepTivator^®^, Miltenyi Biotec, Bisley, UK), representing a structural protein in the ChAdOx viral coat and an immunodominant T-cell target among capsid proteins, containing multiple epitopes conserved among serotypes. Furthermore, 0.45% DMSO was used as a negative control and PHA as a positive control, prepared as 4X solutions in Complete Media. Cryopreserved peripheral blood mononuclear cells (PBMC) were thawed, resuspended in Complete Media, and seeded at 1 × 10^5^ viable cells/well. The plates were incubated for 18 h ± 30 min at 37 °C and 5% CO_2_. PBMC were removed, and the plates were incubated with 1.6 µg/mL of anti-human IFNγ detection antibody (BD Biosciences) for 2 h, followed by horseradish peroxidase-conjugated streptavidin (1:100) (BD Biosciences) for 1 h. The plates were developed for ~10 min using an AEC Substrate Set (BD Biosciences), and spots were counted using an AID (iSpot) spectrum reader. The replicate raw spot counts were normalised to spot-forming units/10^6^ PBMC, values were averaged, and the background was subtracted for data plotting and statistical analysis.

Intracellular cytokine staining using flow cytometry: Intracellular cytokine staining was used to measure the IFNγ, TNFα, IL-2, CD107a, and CD154 produced by the HBV antigen or hexon-specific CD4+ or CD8+ T cells in PBMC across the study’s time points. Cryopreserved PBMC were thawed, resuspended in Complete Media, seeded at 1 × 10^6^ viable cells/well in 96 well V-bottom plates, and rested overnight at 37 °C, 5% CO_2_. PBMC were stimulated with four HBV or one AdV5 hexon peptide pools (as described for the IFNγ ELISpot) or 0.45% DMSO in Complete Media wells as negative controls. Phorbol myristate acetate (PMA) and Ionomycin (Biolegend, London, UK) were added to the positive control, set-up, and unstained control wells. A cocktail of protein inhibitors, Brefeldin A/Monensin (Biolegend), and anti-CD107a antibody were added to all wells, and the plates were incubated for 6 h ± 30 min at 37 °C, 5% CO_2_. The cells were treated with Fc Block (BD Biosciences) + Viability Dye (Biolegend), followed by staining with a cocktail of surface antibodies. The cells were fixed and permeabilised using Cytofix/Cytoperm buffer (BD Biosciences) and then stained for intracellular cytokines.

Fluorescence Minus One (FMOs) and reference controls were generated and applied to the data analysis. The cells were analysed using a Cytek Northern Lights spectral flow cytometer, and the data analysis was performed on FlowJo software (Version 10.8.1). For details of the dyes and antibodies used in the panel, refer to Appendix A.

ChAdOx1-neutralising antibody assay: ChAdOx1-neutralising antibodies (nAb) were measured using an in vitro-based secreted embryonic alkaline phosphatase (SEAP) quasi-quantitative assay. GripTite 293 MSR cells (Invitrogen, MA, USA) were seeded at 9 × 10^4^ cells/well in 96 well plates and incubated for 24 h. Prior to the MSR cell infection, test sera were 3-fold serially diluted, mixed with a fixed concentration of ChAdOx1-SEAP (Barinthus Biotherapeutics, Oxford, UK), and incubated for 1 h at 37 °C and 5% CO_2_. The final serum dilutions were 1:50, 1:150, 1:450, 1:1350, and 1:12,150. ChAdOx1-SEAP was also incubated with Complete Media (DMEM, high glucose, GlutaMAX™ Supplement, pyruvate, 10% Heat-inactivated Foetal Bovine Serum, 600 µg/mL Geneticin) alone (virus control, (VC)). The cells with media alone were plated for background subtraction (cell control (CC)). Post-infection, test plates were incubated for 24 h (5% CO_2_, 37 °C). The following day, supernatants were collected, and the SEAP levels were assayed through the application of CSPD (Invitrogen™ Phospha-Light™ SEAP Reporter Gene Assay System, Fisher Scientific, Loughborough, UK) following the manufacturer’s recommendations. The test sera EC50 values were calculated vs. VC (i.e., the serum titration, which inhibits 50% of the ChAdOx1-SEAP viral infection). EC50 values less than 50 were set to 25 for the statistical calculations.

Statistics: The sample size was based on the ELISpot results of ChAdOx1-HBV administered to healthy adults at a dose of 2.5 × 10^10^, which resulted in a mean of approximately 1000 ± 500 spot-forming units/10^6^ PBMC in three adults who had samples examined by a same-day analysis at Oxford University. Assuming the same response in 15 volunteers in each group, the amendment was powered to detect a 45% decrease in the T-cell response in the participants receiving prior AZD1222 compared to an mRNA-based COVID-19 vaccine. Comparisons of the immune responses used non-parametric statistics (e.g., Mann–Whitney tests for the group comparisons and Wilcoxon signed-rank tests for the paired comparisons within groups). Associations between the immune parameters were assessed by Spearman rank correlations.

## 3. Results

A total of 15 participants out of the 20 screened were enrolled in the PC cohort, but only 11 of the 19 patients with prior Cominarty^®^/Spikevax were able to be enrolled in the PM arm. The vaccines were well-tolerated, with only mild and rare moderate local and systemic reactogenicity in the 7 days following the intramuscular injection (Appendix A). Of note, reactogenicity was less frequent within the PC arm. For example, 40% of the PC participants experienced any systemic reactions versus 81.1% of the PM participants, and 72.7% of the PM participants experienced muscle aches versus 20% of the PC participants.

The IFNγ ELISpot assay was used to quantify the magnitude and breadth of the overall T-cell responses to HBV-specific (Core, Pol1+2, Pol3+4, PreS1/S2+S) and AdV5 hexon peptide pools. The baseline ELISpot responses to the HBV-specific peptide pools were similar between groups (median (IQR): 32.5 (120) vs. 100 (174) SFU/10^6^ PBMC for PC and PM, respectively), with baseline responses primarily to surface antigen, as many of these participants had prior prophylactic HBV vaccination. No participant had evidence of a prior HBV infection, as all had negative hepatitis B core antibodies. The responses over time revealed a greater response on day 14 and day 28 for those in the PM group than the PC group (*p* = 0.070 and 0.013, respectively), but the difference on day 84 had essentially disappeared (*p* = 0.44) (Figure 1).

The breadth of the T-cell response induced by vaccination is defined as the number of positive HBV-specific peptide pools (corresponding to the major HBV regions within the vaccine immunogen (Core, Pol1+2, Pol3+4, PreS1/S2+S)) eliciting a T-cell response. In both the PC and PM cohorts, the breadth of response was higher post-vaccination compared to the baseline. For example, the induction of a T-cell response to Pol1+2 and Pol3+4 post-vaccination was not present on day 0. The increase in the breadth of response was more prominent in the PM cohort (Figure 1C,D).

In contrast to the baseline responses to HBV-specific peptides, which were similar between cohorts, the ELISpot responses to AdV5 Hexon at baseline were greater in the PC cohort than the PM cohort (median (IQR): 87.5 (250) vs. 0 (80) SFU/10^6^ PBMC), although the difference was not significant (*p* = 0.10). The summed HBV-specific responses were correlated with the AdV5 hexon responses on day 14 and day 28. While the correlations were not statistically significant, there was a negative correlation trend at both time points (Figure 2). All Mann–Witney *p*-values calculated for a comparison of the ELISpot responses between cohorts are provided in Appendix A.

The Intracellular Cytokine Staining (ICS) assay was used to quantify the CD4+ and CD8+ T-cell responses through an expression of functional cytokines and activation markers in response to stimulation with HBV-specific peptide pools. The ICS results confirmed the HBV-specific ELISpot responses, and the CD8+ T-cell responses dominated, as is the case with such adenoviral vectors. However, the difference in the CD8+ gamma-interferon responses in favour of PM on day 14 (*p* = 0.013) and day 28 (*p* = 0.008) persisted out to day 84 (*p* = 0.01). A similar trend was seen for the CD4+ T cells (*p* = 0.003 at day 14, *p* = 0.010 at day 28, and *p* = 0.21 at day 84) (Figure 3).

The CD8+ TNFα+ responses increased from baseline in both the PC and PM arms (Figure 4); the increase was marginally greater in the PM cohort versus the PC cohort, but the difference was not statistically significant (fold-change from baseline = 1.5 and 1.136 at day 14 for PM and PC, respectively, *p* = 0.44). Overall, no clear trends were observed with the CD4+ TNFα+ responses, both between the cohorts and post-vaccination (Figure 4). The trend for higher post-vaccination frequencies in the PM cohort was observed with both CD8+ and CD4+ IL-2 expressions; however, overall expression frequencies were low for this cytokine (Appendix A).

CD107a and CD154 were assessed as markers of CD8+ T-cell degranulation and activated CD4+ T cells, respectively. As with the CD8+ IFNγ response, CD8+ CD107α responses also favoured the PM cohort on day 14 (*p* = 0.016) and day 28 (*p* = 0.08). The fold-change from baseline within the PM cohort was 2.6 versus 0.8 in the PC cohort on day 14. As with the CD4+ IFNγ response, significance did not persist on day 84 (*p* = 0.32) (Figure 5). The same trend was also observed for the CD4+ CD154+ responses (Figure 6): day 14, *p* = 0.062; day 28, *p* = 0.014; and day 84, *p* = 0.26. However, a fold-change from baseline within the CD4+ CD154+ responses was lower than the CD8+ CD107a+ responses: 1.75 in the PM cohort versus 0.667 within the PC cohort on day 14, further demonstrating that the CD8+ responses dominate with adenoviral vectors. The T-cell responses were increased post-vaccination in both arms and, to a greater extent, in the PM. All Mann–Witney *p*-values calculated for a comparison of the ICS responses between cohorts are provided in Appendix A.

The ChAdOx1-neutralising antibody assay was used to measure the anti-ChAdOx1 vector-neutralising antibodies in serum samples at baselines day 0 and day 84. The baseline ChAdOx1-neutralising titers were higher in the PC participants compared to the PM (Figure 7A, median (IQR): 848 (1073) and 25 (0), respectively; *p* < 0.0001). Both groups had increases in the adenoviral neutralisation response on day 84 versus the baseline (Figure 7B, median (IQR): 1.3-fold (0.9) versus 6.2-fold (5.2), for PC and PM, respectively (*p* = <0.0001).

As seen in Figure 2, higher baseline adenoviral T-cell responses were observed in the PC cohort compared to the PM, and their relationship to subsequent day 14 and day 28’s ELISpot responses trended towards a negative correlation. The relationship between the baseline neutralisation titers and the magnitude of the HBV-specific ELISpot response on day 14 and day 28 is shown in Figure 8A,B. Although a significant correlation was seen on day 28 for all participants (*p* = 0.027), there was no clear correlation with the peak ELISpot responses for the PC (or PM) group alone (Figure 8C,D).

## 4. Discussion

We demonstrated that, if given within a three-month time frame, a prior immunisation with a homologous adenoviral vector encoding for one insert can impact the subsequent T-cell response induced towards a heterologous insert encoded by the same vector. Previous observations in human adenoviral vaccine trials have shown a clear effect of prior immunity following natural adenoviral infection to immunisation using the same vector and possibly heterologous simian vectors [2,3,4,19,20]. However, there are varying data on the T-cell effect of prior non-replicating adenoviral immunisation and neutralisation on subsequent injections, as the epitopes and magnitude of the effect vary from that of natural infection [10,11,21,22].

In studies of homologous boosting of ChAdOx1 for the prevention of COVID-19 by Vaxzevria or hAd26, there was a minimal correlation between the neutralising antibody titers following the first vaccination to antibody responses following the boost [7,8,13,23,24]. However, it is well established that boosting immune responses requires a reasonable delay of months for adequate improvement in the magnitude of the response [25]. Whether this delay reduces the killing of antigen-presenting cells by CD8+ T-cell responses to the vector or the insert or by neutralisation of the vector is not clear.

In a comprehensive but complicated study in non-human primates primed using hAd26/hAd26, hAd26/hAd35, or hAd26/MVA encoding an RSV antigen and then boosted with hAd26, hAd35, or MVA encoding varying Ebola antigens, minimal interference was observed [26]. There did appear to be an effect in the T-cell response in one previously immunised group at a 4-week time point, which is consistent with our data, although the antibody responses were variable and difficult to interpret. Importantly, the interval to the boost varied from 26 to 55 weeks, with no clear impact by the varying length of the boost delay. Of note, a regimen of Ad26/MVA Ebola antigens followed 57 weeks later by the same regimen encoding HIV antigens showed improved cellular responses with prior priming.

These data are of critical relevance, as the adenoviral COVID-19 vaccines have now been used in hundreds of millions of individuals worldwide. Given the complexity of the data, we had advised that our ChAdOx1 vectors in programs for hepatitis B, cancer, and HPV only be administered with a minimal interval of 3 months from prior adenoviral vaccines, with the widespread use during COVID-19 in mind, but this interval was based on expert opinion, and not derived empirically. The data generated in this study show an impact on the peak HBV antigen-specific T-cell responses from a Vaxzevria^®^ injection given about three months prior, and as a result, we suggest an extended interval.

The data from both the J&J HIV and the Gritstone personalised vaccine cancer trials have shown that an interval over one year is associated with significant T-cell-response boosting for a homologous vector and antigen regimen [13,27]. Whether this can be shortened to 6 months for a heterologous antigen in the same adenoviral vector backbone is yet to be definitively proven. Additionally, this may vary for different antigens, dosing regimens, or heterologous vectors that induce varying innate and adaptive immune responses and use different cellular entry pathways, and therefore require more systematic data or modelling. The lack of attention to modelling and understanding the immune network related to dose and interval is only now being addressed, and hopefully, more studies in this area will shine a light on methods to better address vaccine development.

## 5. Conclusions

To summarise, we have shown that a 3-month interval between initial exposure to ChAdOx1 (ChAdOx1-SARS-CoV2, Vaxzevria^®^) and re-exposure to the same viral vector containing a heterologous antigen insert is not long enough to overcome the interference of anti-vector immunity. To further clarify the optimal interval between first and second doses of ChAdOx1-based products to circumvent the impact of anti-vector immunity, we are planning to evaluate serum samples collected at longer intervals as part of other trials using the ChAdOx1 viral vector platform for the presence of neutralising antibodies, where T-cell responses to antigen inserts have also been assessed. The optimal interval is expected to be between 3 and 12 months, given that published data demonstrate robust T-cell boosts when a second ChAd vector dose is administered between 6 months and 2 years after the first [12,27].

## Figures and Tables

**Figure 1 vaccines-12-00644-f001:**
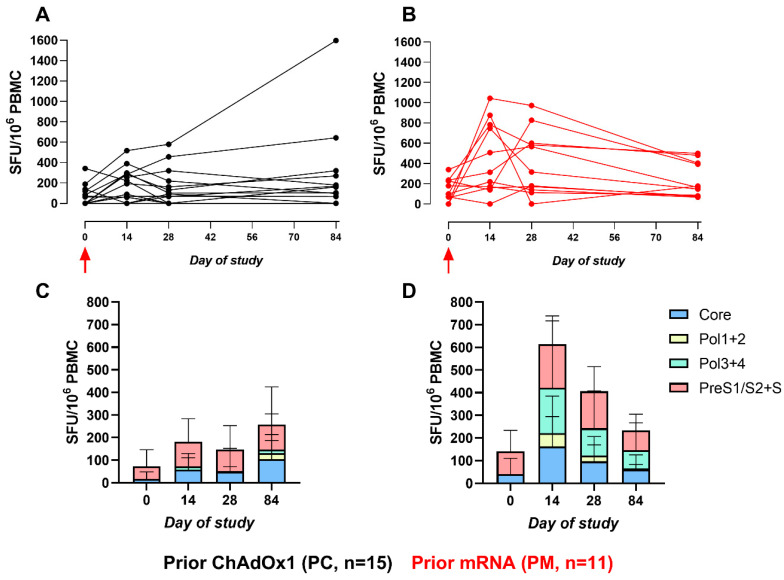
The sum of HBV-specific IFNγ ELISpot responses. (**A**,**B**): Line graphs represent the sum of HBV-specific peptide responses (Core, Pol1+2, Pol3+4, PreS1/S2+S, and SFU/10^6^ PBMC) for each participant across the study for PC and PM, respectively. Red arrow indicates ChAdOx1-HBV administration. (**C**,**D**): Stacked bar graphs represent the sum of mean responses to HBV-specific peptide pools for PC and PM, respectively.

**Figure 2 vaccines-12-00644-f002:**
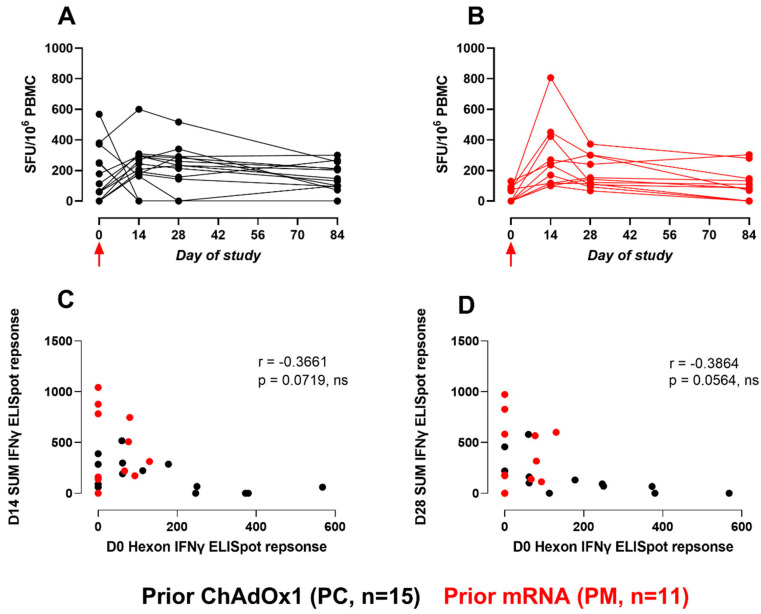
Gamma-interferon ELISpot responses to the AdV5 Hexon peptide pool. (**A**,**B**): Line graphs represent AdV5 Hexon responses (SFU/10^6^ PBMC) for each participant across the study for PC and PM cohorts, respectively. Red arrow indicates ChAdOx1-HBV administration. (**C**,**D**): Correlation of baseline AdV5 Hexon responses to the sum of HBV-specific peptide response (Core, Pol1+2, Pol3+4, and PreS1/S2+S) on day 14 and day 28, respectively (SFU/10^6^ PBMC). Correlations were assessed using Spearman tests.

**Figure 3 vaccines-12-00644-f003:**
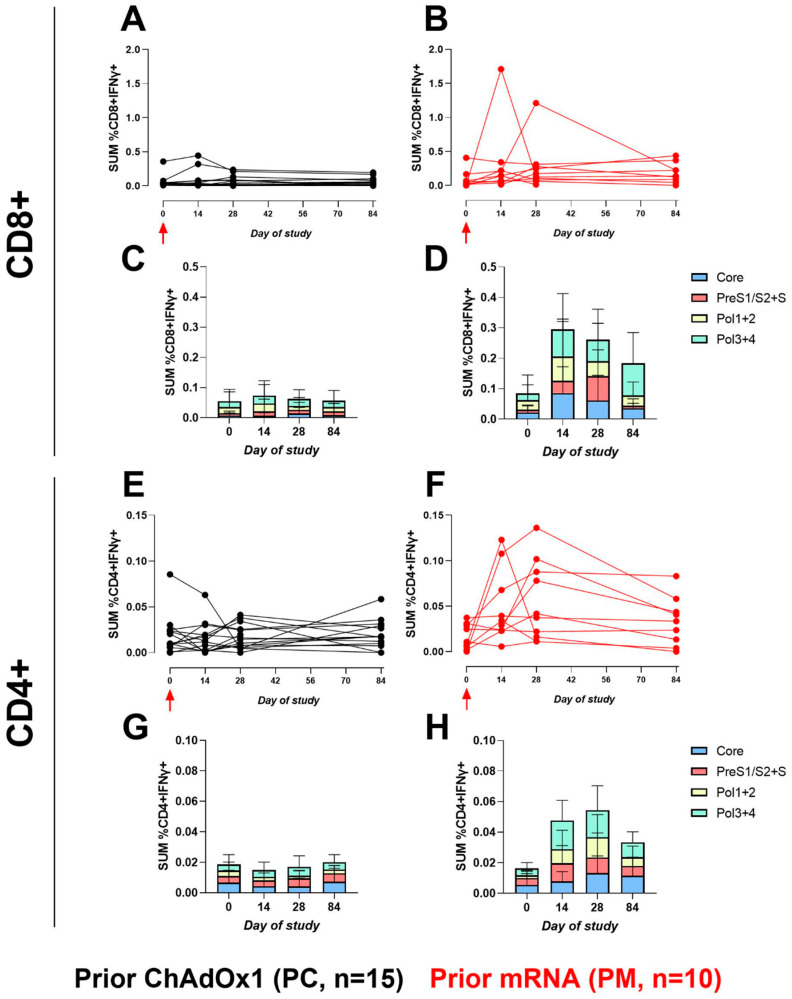
The sum of HBV-specific CD8+ or CD4+ IFNγ+ ICS responses. Line graphs represent the sum of CD8+ or CD4+ IFNγ+ HBV-specific peptide responses (Core, Pol1+2, Pol3+4, PreS1/S2+S, % of total population) for each participant across the study for PC and PM, respectively. Red arrow indicates ChAdOx1-HBV administration. (**A**), CD8+, PC, (**B**), CD8+, PM, (**E**), CD4+, PC, and (**F**), CD4+, PM. Stacked bar graphs represent the sum of mean responses to HBV-specific peptide pools. (**C**), CD8+, PC, (**D**), CD8+, PM, (**G**), CD4+, PC, and (**H**), CD4+, PM. Note: difference in scale of the Y-axes between CD8+ and CD4+ plots.

**Figure 4 vaccines-12-00644-f004:**
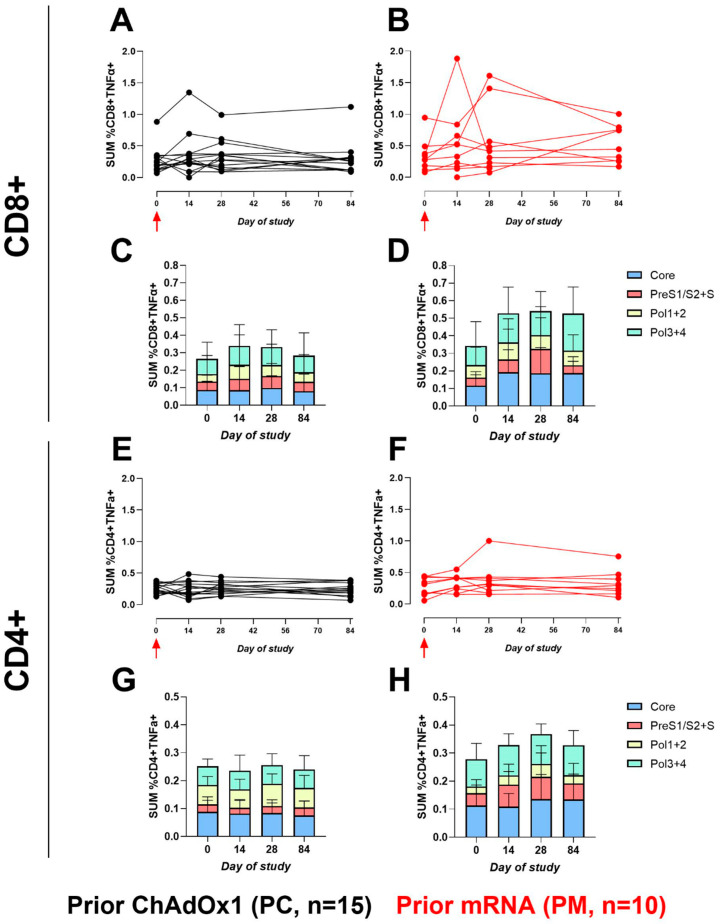
The sum of HBV-specific CD8+ or CD4+ TNFα+ ICS responses. Line graphs represent the sum of CD8+ or CD4+ TNFα+ HBV-specific peptide responses (Core, Pol1+2, Pol3+4, and PreS1/S2+S, % of total population) for each participant across the study for PC and PM, respectively. Red arrow indicates ChAdOx1-HBV administration. (**A**), CD8+, PC, (**B**), CD8+, PM, (**E**), CD4+, PC, and (**F**), CD4+, PM. Stacked bar graphs represent the sum of mean responses to HBV-specific peptide pools. (**C**), CD8+, PC and (**D**), CD8+, PM, (**G**), CD4+, PC, and (**H**), CD4+, PM. Note: difference in scale of the Y-axes between CD8+ and CD4+ stacked bar plots.

**Figure 5 vaccines-12-00644-f005:**
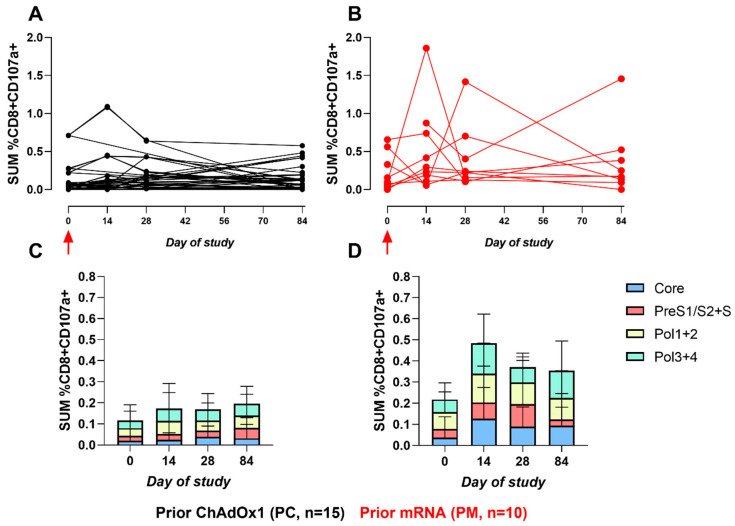
The sum of HBV-specific CD8+ CD107a+ ICS responses. (**A**,**B**): Line graphs represent the sum of CD8+ CD107a+ HBV-specific peptide responses (Core, Pol1+2, Pol3+4, and PreS1/S2+S, % of total CD8+ population) for each participant across the study for PC and PM, respectively. Red arrow indicates ChAdOx1-HBV administration. (**C**,**D**): Stacked bar graphs represent the sum of mean responses to HBV-specific peptide pools for PC and PM, respectively.

**Figure 6 vaccines-12-00644-f006:**
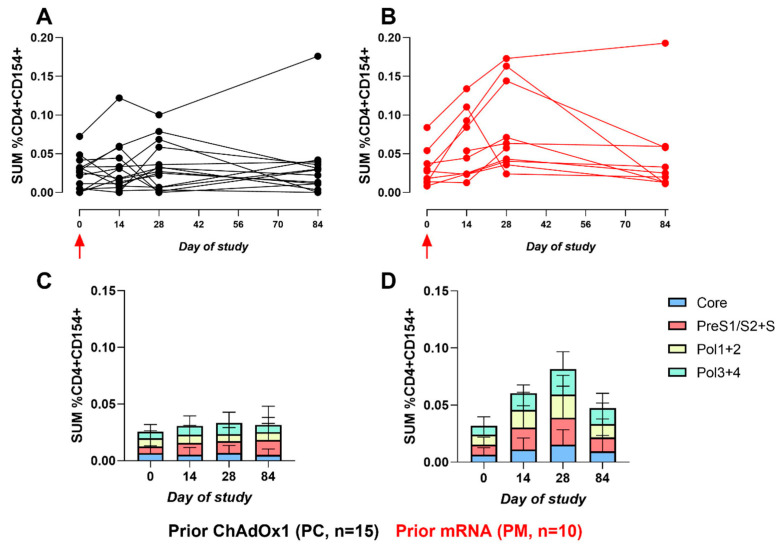
The sum of HBV-specific CD4+ CD154+ ICS responses. (**A**,**B**): Line graphs represent the sum of CD4+ CD154+ HBV-specific peptide responses (Core, Pol1+2, Pol3+4, and PreS1/S2+S, % of total CD4+ population) for each participant across the study for PC and PM, respectively. Red arrow indicates ChAdOx1-HBV administration. (**C**,**D**): Stacked bar graphs represent the sum of mean responses to HBV-specific peptide pools for PC and PM, respectively.

**Figure 7 vaccines-12-00644-f007:**
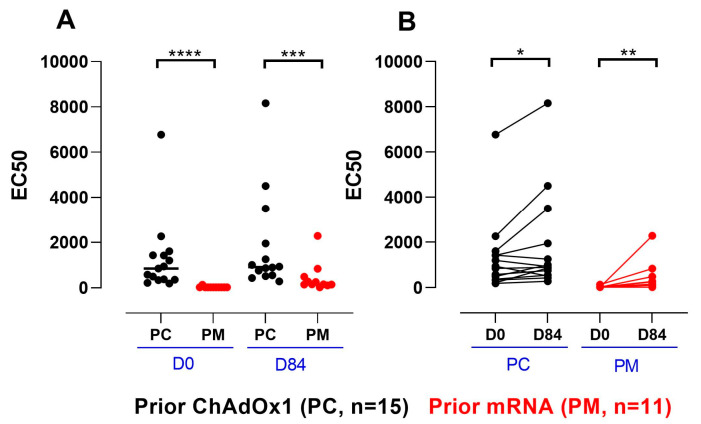
Adenoviral neutralisation titers of sera in PC and PM on day 0 and day 84. (**A**): Neutralising antibody (nAb) titers grouped by time point, compared using a Mann–Whitney test (day 0: *p* < 0.0001 (****); day 84: *p* = 0.0007 (***)). (**B**): nAb titers grouped by cohort, compared using a Wilcoxon signed-rank test for paired data (PC: *p* = 0.034 (*), PM: *p* = 0.002 (**)).

**Figure 8 vaccines-12-00644-f008:**
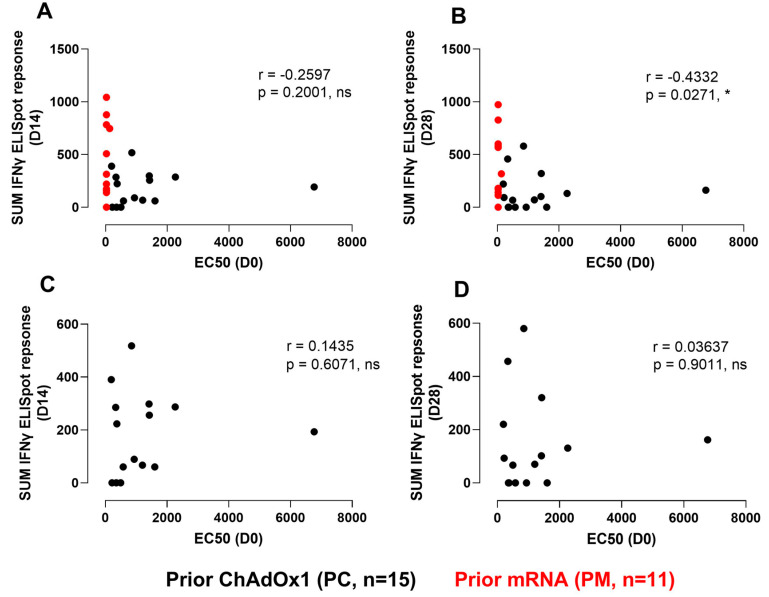
Correlation of baseline (D0) neutralisation titers to peak gamma-interferon ELISpot responses at day 14 (**A**,**C**) and day 28 (**B**,**D**) post immunisation. (**A**,**B**): PC and PM data. (**C**,**D**): PC-only data. SUM IFNγ ELISpot response refers to the sum of HBV-specific peptide responses (Core, Pol1+2, Pol3+4, and PreS1/S2+S). Correlations were assessed using Spearman tests (*, *p* < 0.05). Black dots, PC. Red dots, PM.

## Data Availability

The data presented in this study are available on request from the corresponding author.

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
