# Peer review of "Effect of Prior ChAdOx1 COVID-19 Immunisation on T-Cell Responses to ChAdOx1-HBV†"

_vaccines, 2024, doi:10.3390/vaccines12060644_

Round 1

Reviewer 1 Report

Comments and Suggestions for Authors

            The authors the effect of prior ChAdOx1 COVID-19 immunization on T cell 2 responses to ChAdOx1-HBV. The information is relevant and interesting. Some concerns and limitations should be addressed before publication.

1.     The number of samples analyzed is lacking in the Abstract.

2.     More background on the (therapeutic) vaccine candidate is needed in the Introduction. Previous studies were based on healthy volunteers. Was their HBV vaccination status taken into account in the Phase I study and particularly in this study?

3.     In Results, the authors mentioned that none of the participants had serological evidence of HBV infection, although some of them (how many?) had serological evidence of previous vaccination.

4.     Finally, the number of samples is shown in results, and is somehow low for establishing the comparisons aimed in this study.

5.     Line 197. Please edit the following sentence: ¨the breadth of response was higher post-vaccination compared to baseline ¨.

6.     Figures. No indication on the number of samples analyzed is provided. Was there any significant difference between PC and PM?

7.     Figures. No indication is provided if the samples were from previously HBV vaccinated individuals or not. This parameter may influence the results.

8.     Figure 1C and D. The stacked bar represents the sum of mean responses to HBV specific peptide pools at what time?

9.     Discussion, line 327. It is clear that it is advisable to wait for at least 3 months before re-immunizing with an adenoviral vector. However, what is the evidence that longer time would be even better or that 3 months is enough? This was not tested in this study.

10.  In conclusion, the number of samples analyzed is somehow low, there is no evidence that 3 months would be enough time for reducing the interference, and the authors do not discriminate in their samples which one were from individuals previously with HBV, which undoubtedly may affect the results. These aspects seriously hamper the conclusions of this study, and are not even signaled as limitations in Discussion.

Comments on the Quality of English Language

Justo a minor edition described in the comments to authors.

Reviewer 2 Report

Comments and Suggestions for Authors

I read the manuscript submitted to me for review with great interest. In my opinion, it has the potential to be published but needs some changes.

Abstract: we talk about healthy volunteers, but how many? The purpose of the research is unclear.

Introduction: As with the abstract, the purpose of the research should be more clearly outlined. The references are consistent with the theme of the research.

M & M: Perhaps it would be appropriate to specify the number of cases here, without referring to a previous publication.

Results: nothing to report.

Discussion: Well-articulated. However, a conclusion that concisely summarizes what the research highlights is missing.

References: adequate

Reviewer 3 Report

Comments and Suggestions for Authors

The authors analysed the effect of prior ChAdOx1-SARS-CoV2 immunization on the T cell responses to ChAdOx1 vector encoding HBV antigens in healthy volunteers who had received Vaxzevria or the Pfizer or Moderna mRNA COVID-19 vaccine between 10 to 18 weeks before by ELISpot and ICS. This is an important investigation as adenoviral COVID-19 vaccines have now been used in hundreds of millions of individuals worldwide. The manuscript is well-written and methodology looks correct.

Minor points:

1.     Please insert “Prior ChAdOx1, Prior mRNA” into all Figures as shown in Figure 2 & 8 for easier understanding.

Round 2

Reviewer 1 Report

Comments and Suggestions for Authors

The authors addressed most of the concerns.